# Oral Treatment with RD2RD2 Impedes Development of Motoric Phenotype and Delays Symptom Onset in SOD1^G93A^ Transgenic Mice

**DOI:** 10.3390/ijms22137066

**Published:** 2021-06-30

**Authors:** Julia Post, Anja Schaffrath, Ian Gering, Sonja Hartwig, Stefan Lehr, N. Jon Shah, Karl-Josef Langen, Dieter Willbold, Janine Kutzsche, Antje Willuweit

**Affiliations:** 1Institute of Biological Information Processing, Structural Biochemistry (IBI-7), Forschungszentrum Jülich, 52425 Jülich, Germany; j.post@fz-juelich.de (J.P.); a.schaffrath@fz-juelich.de (A.S.); i.gering@fz-juelich.de (I.G.); 2Institute for Clinical Biochemistry and Pathobiochemistry, German Diabetes Center, Leibniz Center for Diabetes Research at Heinrich-Heine-University Düsseldorf, 40225 Düsseldorf, Germany; sonja.hartwig@ddz.de (S.H.); stefan.lehr@ddz.de (S.L.); 3German Center for Diabetes Research, Partner Düsseldorf, 85764 München-Neuherberg, Germany; 4Institute of Neuroscience and Medicine, Medical Imaging Physics (INM-4), Forschungszentrum Jülich, 52425 Jülich, Germany; n.j.shah@fz-juelich.de (N.J.S.); k.j.langen@fz-juelich.de (K.-J.L.); 5Institute of Neuroscience and Medicine 11, INM-11, JARA, Forschungszentrum Jülich, 52425 Jülich, Germany; 6JARA-Brain-Translational Medicine, 52062 Aachen, Germany; 7Department of Neurology, RWTH Aachen University, 52062 Aachen, Germany; 8Department of Nuclear Medicine, RWTH Aachen University, 52062 Aachen, Germany; 9Institut für Physikalische Biologie, Heinrich-Heine-Universität, 40225 Düsseldorf, Germany

**Keywords:** amyotrophic lateral sclerosis, behaviour, motor coordination, plasma cytokines, d-enantiomeric peptide, neuroinflammation, SOD1^G93A^

## Abstract

Neuroinflammation is a pathological hallmark of several neurodegenerative disorders and plays a key role in the pathogenesis of amyotrophic lateral sclerosis (ALS). It has been implicated as driver of disease progression and is observed in ALS patients, as well as in the transgenic SOD1^G93A^ mouse model. Here, we explore and validate the therapeutic potential of the d-enantiomeric peptide RD2RD2 upon oral administration in SOD1^G93A^ mice. Transgenic mice were treated daily with RD2RD2 or placebo for 10 weeks and phenotype progression was followed with several behavioural tests. At the end of the study, plasma cytokine levels and glia cell markers in brain and spinal cord were analysed. Treatment resulted in a significantly increased performance in behavioural and motor coordination tests and a decelerated neurodegenerative phenotype in RD2RD2-treated SOD1^G93A^ mice. Additionally, we observed retardation of the average disease onset. Treatment of SOD1^G93A^ mice led to significant reduction in glial cell activation and a rescue of neurons. Analysis of plasma revealed normalisation of several cytokines in samples of RD2RD2-treated SOD1^G93A^ mice towards the levels of non-transgenic mice. In conclusion, these findings qualify RD2RD2 to be considered for further development and testing towards a disease modifying ALS treatment.

## 1. Introduction

Amyotrophic lateral sclerosis (ALS) is a fatal neurodegenerative disease characterised by a progressive motor neuron loss in the central nervous systems. Clinically, ALS leads to focal muscular weakness, with atrophy of skeletal muscles up to progressive paralysis and premature death, usually from respiratory failure, generally in 3 to 5 years after diagnosis [1,2,3]. Most of all ALS cases are sporadic (sALS), while about 10% of the cases are inherited by familial ALS (fALS) [4,5]. More than 150 gene mutations have been reported to be potentially involved in ALS, including 27 which are rated disease-relevant based on strong evidence (http://alsod.iop.kcl.ac.uk/als, accession date 20 March 2021). Mutations in the gene of the Cu/Zn-superoxide dismutase 1 (*SOD1*) are a common cause of fALS [6,7] and are identified in up to 3% of sALS cases [8,9,10,11]. Both fALS and sALS produce similar clinical and pathological features, which supports the strategy of using laboratory models of fALS *SOD1* mutations in preclinical studies to understand disease pathogenesis and to identify potential therapeutic targets for both forms of disease [12,13,14,15]. In 1994, a transgenic mouse model was created (tg(SOD1^G93A^)1Gur), which expresses the mutated human *SOD1* gene (*SOD1^G93A^*) [16,17]. These transgenic mice develop many key features of human clinical signs and pathology of human ALS, including motor neuron degeneration, paralysis and shortened life-span [18,19].

Interestingly, the role of neuroinflammation and immune-inflammatory processes in mutant SOD1-mediated ALS was recognised late, but is now established as an important aspect in human cases and transgenic SOD1^G93A^ mice [20,21,22,23,24,25]. The activation of glial cells in the brain stem and spinal cord is a characteristic hallmark of neuroinflammation in ALS. Especially astrogliosis and activated microglia play a role in disease progression and are considered as indicators during advancement of the disease [26,27,28]. In response to glia cell activation, inflammatory mediators are secreted, e.g., cytokines. Classified into two different types (pro- and anti-inflammatory), cytokines are involved in complex signalling cascades. Pro- and anti-inflammatory cytokines are secreted in unequal levels depending on disease stage [29,30]. Although, many potential targets have been identified and new therapeutics were tested in animal studies and clinical trials, to date there is no curative therapy for ALS present [31,32]. Thus far, only symptomatic treatments are available and minor effects on life prolongation have been achieved [33,34].

A relatively new class of promising drug candidates are the all-d-peptides, which consist solely of d-enantiomeric amino acid residues, and which exhibit several advantages including low immunogenicity and high proteolytic stability [35,36,37]. Recently, the excellent safety and tolerability of an all-d-peptide drug candidate could be demonstrated in a clinical phase I trial [38] after demonstrating its preclinical efficacy in several Alzheimer’s disease mouse models [39,40,41]. The drug candidate, called RD2, binds preferentially amyloid beta (Aβ) monomers with nanomolar affinity and stabilizes Aβ in its native conformation [42]. The rational for developing a head-to-tail tandem version of RD2, RD2RD2, was to obtain a bivalent version of RD2 with potentially higher avidity and affinity for polyvalent Aβ assemblies, like Aβ oligomers. However, RD2RD2 was not significantly efficient neither on amyloid load, soluble and insoluble Aβ nor on cognitive deficits [43]. Instead, RD2RD2 demonstrated remarkable anti-inflammatory effects by reducing neuroinflammation in an Alzheimer’s disease mouse model [44].

Based on these results, we intended to characterise the therapeutic potential of RD2RD2 in an additional mouse model linked to neuroinflammation and chose the ALS SOD1^*G93A^ transgenic mouse model. In a previous study with intraperitoneal treatment of 12 weeks old transgenic SOD1^G93A^ mice, RD2RD2 inhibited disease phenotype progression and demonstrated anti-inflammatory effects by reducing neuroinflammation [44].

To further investigate the general therapeutic potential of RD2RD2 especially after oral administration, we initiated a treatment study with oral application of RD2RD2 in SOD1^G93A^ transgenic mice. The efficacy of the treatment was investigated by analysis of disease phenotype in the mice as well as various behavioural and motor coordination tests. Neuroinflammation was assessed in tissue samples of the brain stem and lumbar spinal cord and by analysis of plasma cytokine levels.

## 2. Results

### 2.1. Oral Treatment with RD2RD2 Led to an Amelioration of the SOD1^G93A^ Phenotype

SOD1^G93A^ transgenic mice and their non-transgenic littermates were administered a daily dose per oral (p.o.) of RD2RD2 (50 mg/kg) or placebo from 53 days ± 2 days of age for nine weeks of treatment.

Weight loss is a frequent feature of ALS and therefore, the body weight of mice was recorded beginning prior to baseline measurements. All mice gained body weight during treatment with placebo-treated transgenic mice showing the least gain of body weight, which became most apparent in the last treatment weeks (Figure 1A,B). At the end of the study, placebo-treated SOD1^G93A^ mice weighed significantly less than non-transgenic littermates.

The SmithKline, Harwell, Imperial College, Royal Hospital, Phenotype Assessment (SHIRPA) test battery was used to monitor the progression of the neurodegenerative phenotype of transgenic placebo- or RD2RD2-treated mice. After two weeks of the treatment period, placebo-treated SOD1^G93A^ mice showed significant behavioural impairments compared to their non-transgenic littermates (Figure 1C). One week later, behavioural impairments became significant in the RD2RD2-treated SOD1^G93A^ mice compared to their non-transgenic littermates (Figure 1C). Treatment with RD2RD2 significantly improved the behavioural phenotype of SOD1^G93A^ mice vs. placebo-treated littermates already three weeks after treatment start and thereafter (Figure 1C). While the behavioural deficits of placebo-treated mice progressed steadily during the nine weeks treatment period, RD2RD2 treatment slowed down phenotype progression drastically. For additional investigation of the phenotype progression, parameters of the SHIRPA test were subdivided into a motor score (Figure 1D). After four weeks of treatment, motor deficits were significantly lower upon RD2RD2- vs. placebo-treated SOD1^*G93A^ mice (Figure 1D).

### 2.2. RD2RD2 Treatment Improved Motor Performance and Led to Delay of the Disease Onset in SOD1^G93A^ Mice

To measure motor deficits during disease progression, additional motor tests were carried out, i.e., pole test, grip strength and splay reflex test of hind limbs (Figure 2A–C). These tests detect functional deficits in early pathological motor defects in SOD1^G93A^ mice. Throughout the experiment, non-transgenic mice did not show any motor deficits, only slight but normal variability in their daily performance during this period. In contrast, transgenic SOD1^G93A^ mice developed significant deficits during the study on motor performance at different time-points.

As early as three weeks after treatment start, RD2RD2-treated SOD1^G93A^ mice showed significantly less impairment in pole performance and splay reflex test than placebo-treated littermates. Moreover, the efficacy of RD2RD2 in SOD1^G93A^ mice vs. placebo-treated littermates was demonstrated starting after five weeks of treatment in the grip strength test.

Assessment of disease onset according to Mead et al. (2011) [45], i.e., defect in hind limb splay and tremor, was performed three times a week from the start of the experiment. First disease onset occurred at 86 days of age in placebo-treated mice (Figure 2D). Kaplan-Meier survival analysis revealed that the placebo group had a mean disease onset of 95 days (*n* = 14, range between 86 days and 103 days), whereas RD2RD2-treated SOD1^G93A^ mice lived on average 106 days (*n* = 13, range between 94 days and 113 days) without any symptoms. Log rank test was statistically significant (^###^ *p* < 0.001), indicating a mean delay of disease symptoms of 11 days by RD2RD2 treatment in SOD1^G93A^ mice. One RD2RD2-treated SOD1^G93A^ mouse showed no tremor at all during the course of the treatment study (Figure 2D). Evidence of a changed hind limb splay or tremor was undetected at any stage in non-transgenic littermates. In general, RD2RD2-treated SOD1^G93A^ mice showed improved motor performance in comparison to placebo-treated littermates. All non-transgenic mice exhibited normal motor function throughout the experimental period.

### 2.3. Reduced Gliosis and Neurodegeneration in RD2RD2-Treated SOD1^G93A^ Mice

During disease progression, SOD1^G93A^ mice are pathologically characterised by the presence of gliosis and neurodegeneration in brain and spinal cord, especially within the brain stem, motor cortex and the lumbar spinal cord [46,47,48,49]. Histopathological quantifications were carried out after nine weeks of oral treatment.

In general, quantification revealed less microgliosis in SOD1^G93A^ mice vs. non-transgenic littermates (Figure 3A,B and Appendix A). RD2RD2 treatment diminished the activation of microglia stained with antibody CD11b in the brain stem of SOD1^G93A^ mice in comparison to placebo-treated mice. The same trend was observed in the lumbar spinal cord, which did not reach statistical significance (Figure 3A,B and Appendix A). In addition, the amount of GFAP-positive cells in the brain stem of RD2RD2-treated SOD1^G93A^ mice was significantly less than in the placebo group (Figure 3C,D and Appendix A).

To support the finding that RD2RD2-improved motor performance and delayed disease onset, we analysed the number of neuronal cells in the brain of SOD1^G93A^ mice. Neuronal changes were quantified by the number of NeuN-positive neurons in the brain stem and motor cortex of SOD1^G93A^ mice and their non-transgenic littermates. Additionally, lower motor neurons of the brain stem were assessed by quantification with a ChAT-specific antibody. Placebo-treated mice displayed a significant loss of mature neurons in the brain stem and motor cortex (Figure 3E,F and Appendix A), and in brain stem specifically a significant decrease of lower motor neurons in comparison to non-transgenic littermates (Figure 3G,H and Appendix A). Treatment with RD2RD2 rescued the loss of total neurons in both the brain stem and motor cortex of SOD1^G93A^ mice to levels of non-transgenic littermates and, moreover, of motor neurons in the brain stem of RD2RD2-treated SOD1^G93A^ mice (Figure 3E–H and Appendix A).

In order to assess whether there is a possible relationship between phenotype progression and levels of neuroinflammation or neurodegeneration, results of histological quantifications were correlated with both, the SHIRPA score and the SHIRPA motor score of the last treatment week. The Pearson’s correlation coefficients from each correlation are shown in Appendix A.

Analysis over all three groups revealed a significant positive correlation of neuroinflammation with the SHIRPA score and, moreover, with the SHIRPA motor score (Figure 4A–D and Appendix A, overall effect of ntg vs. placebo SOD1^G93A^ vs. RD2RD2 SOD1^G93A^). The number of mature neurons correlated significantly negative with the SHIRPA scores over all three groups (Figure 4E,F and Appendix A) and specifically with the SHIRPA motor score (Figure 4F and Appendix A).

### 2.4. RD2RD2 Administration Suppressed the Activation of Inflammatory Plasma Markers in SOD1^G93A^ Mice

To identify whether RD2RD2 modulates the general inflammatory status in transgenic SOD1^G93A^ mice, a multiplex immunoassay was used to analyse a possible change of plasma cytokines at the end of the study. Analysis of the inflammatory marker levels revealed a significant up- or downregulation in several cytokines of placebo-treated SOD1^G93A^ mice vs. non-transgenic littermates (Figure 5 and Appendix A; IL-1β, IL-6, IL-10, IL-12p40, IL-13, IL-17, CCL-2 and CXCL-1). The RD2RD2-treated group lost significant changes of inflammatory markers in comparison to the non-transgenic (Figure 5A,C,D,I and Appendix A; IL-1β, IL-6, IL-10 and CCL-2) and gained differences to placebo-treated mice (Appendix A). Generally, there was a trend of reversing the altered plasma levels of SOD1^G93A^ mice towards levels of non-transgenic littermates, reaching statistical significance in the following cytokines: IL-10, IL-13 and CCL-2.

## 3. Discussion

ALS is one of the most common, fatal and progressive neurodegenerative disease affecting the motor system [50]. Due to an ageing world population, there is an increasing incidence of ALS cases during the last decades [51]. Despite intensive research, the cause is still unknown and current treatment options are only symptomatic [34]. In addition to the clinical symptoms like motor deficits, an increasing evidence has firmly certified that the dysregulation of the immune system in ALS results in an extensive inflammatory response [52,53]. Previous studies demonstrated the key role of neuroinflammation, mainly caused by microglia activation and astrogliosis [27,54,55]. In the early phase of disease, activated glial cells produce a protective immune response, while during the progression of the disease excessive glial expression is harmful and resulted in a neurotoxic response [23,56]. Further, glia cell activation leads to changes in the production and release of inflammatory markers [25,57,58].

In the hereby described work, we examined the compound RD2RD2 for its therapeutic efficacy in the SOD1^G93A^ mouse line, a model for the neurodegenerative, neuroinflammation-driven disease ALS. RD2RD2 has been discovered in a screening campaign against Aβ and demonstrated remarkable anti-inflammatory effects by reducing neuroinflammation in an Alzheimer’s disease mouse model [44]. Based on these findings, an in vivo study with RD2RD2 in an ALS specific mouse model was initiated to explore and validate the therapeutic potential of RD2RD2 when given orally to mice.

Seven weeks old SOD1^G93A^ mice and their non-transgenic littermates were treated daily with RD2RD2 or placebo, formulated in tailor-made jellies. Oral treatment was chosen as the route of administration because it is the least invasive application method and preferred route of administration in humans. The early stage of ALS in SOD1^G93A^ mice is characterised by neuroinflammation exhibiting a neuroprotective function. Progression of disease leads to a conversion into a neurotoxic reaction, which accelerates the process [59,60]. Several research groups demonstrated time-dependent effects on neuroinflammation and disease progression in SOD1^G93A^ mice upon treatment with anti-inflammatory compounds [61,62,63]. Therefore, the purpose of the current study was to investigate the therapeutic potential on ALS pathogenesis, when initial treatment starts in the early phase of ALS before symptom onset.

In this study, we were able to demonstrate an enhanced in vivo efficacy of RD2RD2. Placebo-treated SOD1^G93A^ mice showed a significant progression of the motor-neurodegenerative phenotype while RD2RD2 treatment in transgenic mice slowed down phenotype progression, especially during the first treatment weeks. Consistent with the results of the behavioural and motor coordination tests, RD2RD2-treatment significantly delayed disease onset in SOD1^G93A^ mice. Thus, an early treatment was able to extend the time-span to disease onset between placebo- and RD2RD2-treated SOD1^G93A^ mice. Till the end of the behavioural experiments, the difference in phenotype and motor performance of RD2RD2- vs. placebo-treated SOD1^G93A^ mice were significant.

Immunohistochemical investigations of gliosis in the brain stem and lumbar spinal cord resulted in significant differences between transgenic and non-transgenic mice and correlated significantly with the neurodegenerative phenotype in the last treatment week. However, gliosis in the brain stem of RD2RD2-treated SOD1^G93A^ mice was lower than in placebo-treated littermates with a statistically significant difference. The data suggested that the treatment with RD2RD2 could significantly prevent accumulation of activated glia cells in the brain stem of SOD1^G93A^ mice.

A defining feature of ALS is the fatal progression of neurodegeneration [54,64]. Therefore, we stained mature neurons in the brain stem and motor cortex of all mice and revealed a significant increase in the survival of neurons in RD2RD2-treated SOD1^G93A^ mice vs. the placebo-treated group. The density of mature neurons after RD2RD2 treatment was close to the level of non-transgenic littermates, while neurodegeneration was significantly progressed in the brain stem and motor cortex of the placebo-treated mice. Based on these results, we assume that RD2RD2 has a neuroprotective effect in this mouse model of ALS not only on mature neurons, but also on the survival of motor neurons.

Additionally, plasma samples were examined with a multiplex immunoassay to figure out, whether RD2RD2 treatment had an effect on inflammatory markers, such as cytokines and chemokines. Indeed, RD2RD2 administration normalised the levels of many inflammatory plasma markers in SOD1^G93A^ mice towards levels found in non-transgenic mice. Inflammatory markers reflect disease progression in human cases and the SOD1 mice of ALS and have important roles in both toxic and neuroprotective functions depending on the stage of disease progression [29,30,65,66].

A limitation of the study is certainly the use of only female mice as gender differences are described for sporadic and familial ALS patients [67] as well as for human SOD1 mice [68] and could therefore possibly have an impact on the variability of the results, as it is already described for SOD1^G93A^ mice on an C57BL/6 background [45]. Accordingly, we can only speculate whether the drug is able to elicit the same therapeutic response in male mice, but it might have even stronger effects there. Another limitation is the choice of the mouse model. The SOD1^G93A^ mouse line, although it is one of the most characterised and most commonly used ALS mouse models, is a model which mimics familial ALS based on mutations in the SOD1 gene, which accounts for only a small subset of all ALS cases.

Considering the pathogenic mechanism of ALS, RD2RD2 might be a good therapeutic candidate for a disease modifying treatment in ALS. However, the exact mechanism of action for RD2RD2 has not been elucidated yet. The anti-neuroinflammatory effect of RD2RD2 was a coincidental finding as the compound was originally developed for an anti-Aβ-oligomer directed treatment strategy against Alzheimer’s disease [43]. SOD1^G93A^ mice only secrete endogenous murine Aβ, which is not capable of forming aggregates [69]. Therefore, an Aβ-related mechanism of action for RD2RD2 is unlikely. Instead, an off-target effect of RD2RD2 on a so far unknown target is suggested. Although neuroinflammation cannot trigger ALS per se, activated central nervous system microglia and astrocytes, and immune cells of the periphery, including the immune-modulating cytokines they release, are drivers of disease progression. Not only activated microglia but also astroglia have been shown to exert neurotoxic effects and induce ALS-like symptoms [70]. Accordingly, a direct effect of RD2RD2 on neuroinflammatory cells is able to explain its therapeutic efficacy in this study. The underlying molecular mechanisms, however, need to be investigated in more detail in the future.

## 4. Conclusions

In the hereby described work we examined the all-d-enantiomeric peptide RD2RD2 for its therapeutic potential in an ALS mouse model. So far, the mechanism underlying and the direct target of RD2RD2 are still unknown. Despite this, we were able to demonstrate that prolonged oral treatment with RD2RD2 significantly increased its therapeutic efficacy in SOD1^G93A^ mice. Treatment with RD2RD2 improved the phenotype of SOD1^G93A^ mice in comparison to placebo, led to a reduction of the inflammatory response and decreased neurodegeneration. RD2RD2 is a promising candidate for further development and testing towards a disease modifying treatment in ALS.

## 5. Methods

### 5.1. Animals

Male mice of the congenic B6.Cg-Tg(SOD1^*G93A^)1Gur/J line, which was backcrossed for at least 10 generations to C57Bl/6J, were purchased from JAX (Stock No. 004435, The Jackson Laboratory, Bar Harbor, ME, USA). Male mice were bred in-house with C57BL/6J females obtained from CRIVER (Charles River Laboratories, Sulzfeld, Germany). Mating generated hemizygous SOD1^G93A^ transgenic mice and non-transgenic littermate (ntg) controls. Progenies were genotyped for presence of the human SOD1 gene by a quantitative PCR assay of DNA obtained from ear markings, as previously described [71]. Copy numbers of the transgene were checked by calculation of the delta cycle threshold (∆CT) = CT_internal control_ − CT_gene of interest_. Female SOD1^G93A^ mice with a high copy number of the transgene were selected for stratified randomisation into equal groups. As control, the genotype of each animal was confirmed by a second quantitative PCR at the end of the study. Housing of the animals was under the same terms at the animal facility of the Forschungszentrum Jülich as described previously [72]. Mice were housed in mixed-genotype in a controlled environment (12/12 h light/dark cycle, humidity maintained around 50% and a room temperature between 20 °C and 23 °C). Food and water were available ad libitum.

### 5.2. Drug Candidate

RD2RD2 was purchased from Cambridge Peptides (Cambridge Peptides, Birmingham, United Kingdom) as lyophilized powder. The peptide consists of 24 d-enantiomeric amino acid residues with its C-terminus being amidated (sequence: ptlhthnrrrrrptlhthnrrrrr, 3.2 kDa) (Figure 6).

### 5.3. Study Design

Female SOD1^G93A^ mice were treated daily p.o. with 50 mg/kg RD2RD2 or placebo (drinking water) formulated in tailor-made jellies as described in the section below. The RD2RD2 dosage was chosen based on successful former experiments with related compounds, taking the difference in molecular weight into account [41]. The RD2RD2 amount in the jellies was adjusted based on the weekly determined average body weights of the mice. The cohort of animals contained 42 female mice assigned into three experimental groups consisting of SOD1^G93A^ mice (RD2RD2 *n* = 14 and placebo *n* = 14) and a control group of non-transgenic littermates (ntg *n* = 14). All animals born within one week were stratified into the treatment groups according to gene copy numbers and results of behavioural tests (SHIRPA test and pole test) at baseline. Each mouse was treated for 68 days and the average age at treatment initiation was 53 days ± 2 days.

The anticipated treatment start was based on a previous pilot study in which transgenic SOD1^*G93A^ mice were compared to their wild type littermates from week 4 until week 20 of age. Using the modified pole test, first measurable deficits in motor performance of transgenic mice were detected at an age of 8 weeks [44]. According to the criteria of Ohgomori et al. (2017) [71], the current study was designed to start in the pre-symptomatic phase, before first deficits are visible (7 weeks of age).

In regular intervals, SOD1^G93A^ mice and their non-transgenic littermates were tested in different behavioural set ups (SHIRPA test battery, pole test, grip strength test and splay reflex test of hind limbs) starting before treatment (baseline measurements). The stratified randomisation of the transgenic mice into two equal groups was based on the following criteria: ∆CT and baseline measurement of the SHIRPA and pole test. All tests were carried out at the same time of the day and all mice were habituated in single cages before starting the particular tests. Mice were observed daily for disease progression. At the end of the study, blood and tissue samples were collected of the SOD1^G93A^ and non-transgenic mice. Blood samples were centrifuged, and the supernatant (plasma) was stored at −80 °C. Following blood collections, brains and spinal cords of SOD1^G93A^ and non-transgenic mice were harvested and histologically analysed for gliosis (using an integrin α-M/β-2 (CD11b) antibody for activated microglia and glial fibrillary acidic protein (GFAP) for activated astrocytes) and neurodegeneration (using an antibody against neuronal nuclear proteins (NeuN) for mature neurons and a choline acetyltransferase (ChAT) antibody for motor neurons). Plasma samples were biochemically analysed for several inflammatory markers using a multiplex immunoassay.

### 5.4. Treatment

Jellies used for oral treatment consisted of 19% instant gelatin (Dr. Oetker, Bielefeld, Germany), 30% sucrose and 10% sucralose. To produce drops, the components were solved in the regular drinking water of the mice. In order to produce jellies of suitable size, a 96-well plate was filled with the mixture. Before the gelatin solidified, 50 µL of RD2RD2-solution or drinking water (placebo) was added to each well. The total volume of one jelly was 200 µL. Jellies were stored at 4 °C until further use on the next days. For feeding, mice were placed individually in a clean cage with a jelly. In general, the mice ate the drop within a few minutes. The experimenter ensured that the jelly was eaten before placing the mice back in their home cage. Mice were trained to eat the jelly in preparation of the experiment. The stability of RD2RD2 in jellies was demonstrated by analysing the concentration of RD2RD2 in freshly formulated and 6-day-old jellies (Appendix A). Mice were fed with the same batch of freshly produced jellies for a maximum of 6 days.

### 5.5. Body Weight of SOD1^G93A^

The weight of the SOD1^G93A^ animals was recorded at least three times per week beginning prior to baseline measurements. At disease onset, the animals´ body weight was controlled daily. Body weight was taken always prior to treatment between 7 a.m. and 8 a.m. to avoid diurnal variations.

### 5.6. SHIRPA Phenotype Assessment

The SHIRPA primary screen comprises a test battery of reflex and sensorimotor tests for behavioural assessment of mouse phenotype [73,74]. This test consisted of several subtests: restlessness, alertness, startle response, pinna reflex, corneal reflex, touch response, pain response, grooming, and apathy, abnormal body carriage, abnormal gait, loss of righting reflex, forelimb placing reflex, hanging behaviour, hind limb tremor. Assessment of each mouse began with observation of undisturbed behaviour in habitual environment followed by testing the motor abilities in an arena of 42.5 cm × 18.0 cm × 26.5 cm (L × H × W). Lastly, a sequence of manipulations were used to measure body tone and reflexes. Scoring was defined from 0 (similar to ntg littermates) to 3 (extremely abnormal from ntg littermates). The sum of scores of all subtests per animal was used for analysis. Especially for this mouse model, the last seven tests mentioned above are additionally summed up to a motor score.

### 5.7. Modified Pole Test

Here, a slightly modified version of the standard pole test [75,76] was used to detect early changes in the motor performance of the SOD1^G93A^ mice. The following modifications were realised: Mice were placed head downwards instead of upwards on a vertical pole (height 50 cm, diameter 1.2 cm, rough-surfaced) and their movement downwards was rated (0 continuous run, 1 part-way runs, 2 slipping downwards and 3 falling down). Each animal was tested three times and the sum of all three scores was used for analysis.

### 5.8. Grip Strength Test

Grip strength analysis of the hind limbs was evaluated using a Grip Strength Meter (Ugo Basile Srl, Comerio VA, Italy). Following the manufacturers protocol, mice were placed with all four limbs to a blind top grasping grid for hind limb measurements. The mice were pulled backwards by the tail, at which point they gripped intuitively after the grid. Because of the blind top in the front, only the hind limb force was measured. The peak force was calculated three times in succession. The mean value of all three peak forces was used for analysis.

### 5.9. Splay Reflex Test of Hind Limbs

To characterise motor deficits, mice were suspended up by the tail and the extent of hind limb splaying was assessed for 15 s. A healthy splay of both hind limbs similar to that observed in non-transgenic mice was given a score of 0. An acute splay angle or “weak splay” of both hind limbs received a score of 1. A single leg splay was assigned a score of 2. A mouse that exhibited no splay or pulled both hind limbs together, effectively crossing one over the other, was given a score of 3. This procedure was performed three times and the sum of all three scores was used for analysis.

### 5.10. Disease Onset Analysis

To monitor the disease progression of the SOD1^G93A^ mice, all animals were inspected daily for signs of motor deficits. Disease onset was determined after Mead et al. [45] if the following criteria were both met: “Point at which defects in hind limb splay and enhanced tremor were observed with a score of at least 1 in each category”.

### 5.11. Plasma and Tissue Collection

Mice were deeply anaesthetised by inhalation with isoflurane (1.5–3.0 vol.% in oxygen) and monitored for loss of reflexes in which all the responses to external stimuli cease (verified by a toe pinch). The final collection of blood was done by terminal cardiac puncture. All blood samples were collected in K_2_EDTA tubes and centrifuged for 10 min at 2000× *g* in a cooled centrifuge. The supernatant (plasma) was transferred into pre-labelled tubes. Following blood collections, brains and spinal cords were removed. Brain samples were divided into the two hemispheres and spinal cords transversally before both hemispheres and lumbar spinal cord samples were snap frozen in isopentane. All samples were stored at −80 °C.

### 5.12. Immunohistochemistry

The left-brain hemisphere and the lumbar spinal cord (L1-L5 tract) were cut for immunohistological analysis using a cryotome (Leica Biosystems Nussloch GmbH, Wetzlar, Germany). Immunohistochemistry (IHC) was performed on 20 µm sagittal frozen brain sections and on 12 µm transverse lumbar spinal cord sections (L1-L5 tract). The brain and lumbar region of the spinal cord were identified as described previously [77,78,79]. To avoid differences in staining intensity, which might affect measurements, every staining session was performed in one batch. Gliosis of astrocytes and microglia (antibodies GFAP and CD11b) was assessed in SOD1^G93A^ mice and their non-transgenic littermates. Furthermore, SOD1^G93A^ mice were investigated for mature neurons (antibody NeuN) and motor neurons (antibody ChAT) in the brain. Tissue sections were thawed and fixed with 4 °C pre-cooled 4% paraformaldehyde for 10 min and followed by an antigen retrieval with 70% formic acid for 10 min. The sections were rinsed, and elimination of endogenous peroxidases was ensured by incubation in 3% H_2_O_2_ in methanol for 15 min. After a further washing step, sections were incubated with the primary antibody overnight at 4 °C in a humid chamber (GFAP: DAKO Agilent Technologies, Santa Clara, CA, USA; NeuN: Merck Millipore, Darmstadt, Germany; ChAT: Novus Biologicals Europe, Abingdon, United Kingdom) or for two hours at room temperature (CD11b: Abcam, Cambridge, United Kingdom). Primary antibodies were diluted 1:1000 in tris buffered saline with 1% Triton X-100 (TBST) with 1% bovine serum albumin (BSA) (GFAP and NeuN), 1:2000 in tris buffered saline (TBS) with 1% BSA (CD11b) or 1:1000 in phosphate buffered saline with 0.25% Triton X-100 (PBST) with 1% BSA (ChAT). Afterward, sections of SOD1^G93A^ mice were rinsed and incubated with biotinylated secondary anti-mouse, anti-rabbit or anti-donkey antibody (1:1000 in TBST with 1% BSA (NeuN and GFAP), 1:1000 in TBS with 1% BSA (CD11b) or in PBST with 1% BSA (ChAT), Sigma Aldrich, Taufkirchen, Germany) for two hours at room temperature followed by 3,3′-Diaminobenzidine (DAB) enhanced with saturated nickel ammonium sulfate solution. Immunohistochemical sections were washed in an ascending alcohol series (in following order 70% EtOH, 90% EtOH and 100% EtOH for 5 min) and mounted with DPX Mountant medium (Sigma Aldrich, Taufkirchen, Germany).

### 5.13. Quantification

Images of SOD1^G93A^ sections were taken with a LMD6000 microscope (Leica Microsystems GmbH, Wetzlar, Germany) and the respective software (LAS 4.0 software). All slides were acquired in one microscopy session. Quantification was performed with ImageJ 1.48v (National Institute of Health, Bethesda, MD, USA) and CellProfiler Analyst 1.0415v (Broad Institute, Boston, MA, USA) [80]. Immunoreactive microglial cells (antibody CD11b) and astrogliosis (antibody GFAP) were determined as percentage area (%) of the neuropil occupied by GFAP or CD11b and neuronal nuclei (antibody NeuN) and motor neurons (antibody ChAT) as count per stained area. To avoid deviations in the analysis of the region of interest, a standard circle or rectangle was created with the ImageJ program. CD11b immunoreactive area was analysed in the brain stem and lumbar spinal cord (ntg *n* = 10 to 14, placebo *n* = 9 to 14, RD2RD2 *n* = 10 to 14). GFAP immunoreactive area was analysed in the brain stem and lumbar spinal cord (ntg *n* = 10 to 12, placebo *n* = 10 to 14, RD2RD2 *n* = 10 to 13). NeuN counts were analysed in the brain stem and motor cortex layers 2/3 and 5 (ntg *n* = 10, placebo *n* = 10 to 14, RD2RD2 *n* = 9 to 14). ChAT counts were analysed in the brain stem and motor cortex layers 2/3 and 5 (ntg *n* = 11, placebo *n* = 11, RD2RD2 *n* = 11). In general, 6 brain slices and 8 lumbar spinal cord slices per mouse were used for each quantification. Animals with three or less quantifiable brain slices were excluded from analysis, as well as animals with four or less quantifiable spinal cord samples.

### 5.14. Cytokine Assay

Plasma samples were measured using a multiplex immunoassay (Bio-Plex Pro Mouse Cytokine, Chemokine, and Growth Factor Assays, 23-plex, Bio-Rad Laboratories Inc., Hercules, CA, USA). The measurement of the plasma samples of transgenic SOD1^G93A^ mice (placebo *n* = 11 and RD2RD2 *n* = 11) and their non-transgenic littermates (ntg *n* = 7) was performed according to manufacturer’s protocol. In general, values below the limit of detection (LoD) were excluded from analysis. The plasma samples were tested for several interleukins (IL-1β, IL-4, IL-6, IL-10, IL-12p40 and IL-17), interferon-γ (INF-γ), tumor necrosis factor-α (TNF-α) and several chemokines (C-C motif chemokine ligand (CCL-2 and CCL-5) and C-X-C motif chemokine ligand (CXCL-1). Data corresponded to the concentration ranges of the multiplex immunoassay. Cytokine concentrations are given in picogram per millilitre (pg/mL).

### 5.15. Statistics

All statistical calculations were performed using InVivoStat (Version 3.4.0.0, United Kingdom) or SigmaPlot (Systat Software, v11, Frankfurt/Main, Germany). GraphPad Prism v8 (GraphPad Software Inc., San Diego, CA, USA) was used for the graphic illustrations. Descriptive statistical analyses were calculated on all evaluated parameters. The data are given as means ± SEM. A *p*-value > 0.05 was considered to be not statistically significant (ns). Normal distribution of data was either tested by use of Shapiro–Wilk normality test or by use of a normal probability plot (InVivoStat, Version 3.4.0.0, United Kingdom) [81]. Two-way repeated measurement (RM) ANOVA with Fisher’s Least Significant Difference (LSD) post hoc analysis was used to analyse the results of the behavioural tests. One-way measurement ANOVA with Fisher’s Least Significant Difference (LSD) post hoc analysis was used to analyse the results of the histochemical analysis and biochemical analysis. Statistical analysis of disease onset was performed using the Kaplan–Meier method with log rank test. The correlation between phenotype progression and levels of neuroinflammation or neurodegeneration were measured using Pearson’s correlation coefficient.

## Figures and Tables

**Figure 1 ijms-22-07066-f001:**
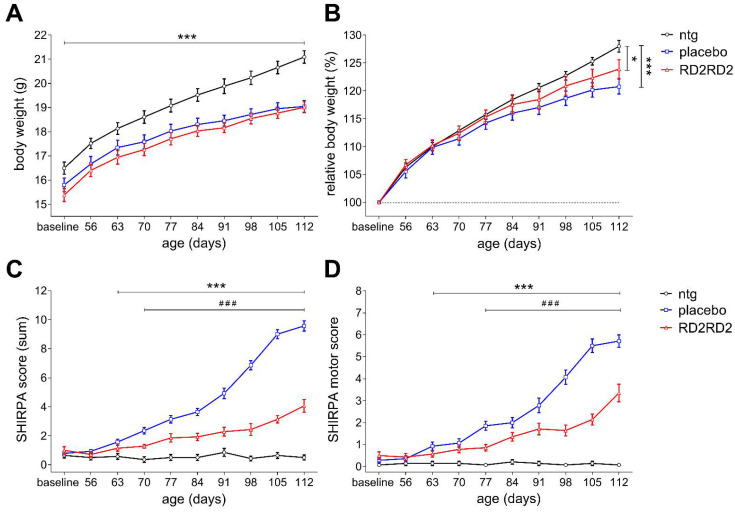
Development of body weight and phenotype in the oral treatment study. Changes in body weight of transgenic and non-transgenic (ntg) mice was analysed over time during treatment (**A**,**B**). Phenotype assessment was performed using the SHIRPA test battery (**C**). Subdivision of the SHIRPA parameters into a motor score revealed additional information of motor symptom progression in RD2RD2- vs. placebo-treated SOD1^G93A^ (**D**). Of note, the higher the score the worse the performance of the mice in the analyses in C and D. Data is represented as mean ± SEM. Statistical calculations were conducted by two-way RM ANOVA with Fisher’s LSD post hoc analysis, *n* = 14 each group (**A**–**D**). Results of analysis: (**A**) absolute body weight (g), F(2,351) = 10.76, *p* < 0.001, Fisher LSD post hoc analysis, ntg vs. placebo *p* < 0.001, ntg vs. RD2RD2 *p* < 0.001; (**B**) relative body weight (%), F(18,351) = 3.84, *p* < 0.001, Fisher LSD post hoc analysis, ntg vs. placebo *p* < 0.001, ntg vs. RD2RD2 *p* = 0.013; (**C**) SHIRPA test, F(2,351) = 255.58, *p* < 0.001, Fisher LSD post hoc analysis, ntg vs. placebo *p* < 0.001, ntg vs. RD2RD2 *p* < 0.001 and F(1,234) = 146.26, *p* < 0.001, Fisher LSD post hoc analysis, placebo vs. RD2RD2 *p* < 0.001 and (**D**) SHIRPA motor test, F(1,234) = 53.39, *p* < 0.001, Fisher’s LSD post hoc analysis, placebo vs. RD2RD2 *p* < 0.001. Asterisks (*) indicate significance between non-transgenic and placebo group (ntg vs. placebo or ntg vs. RD2RD2: * *p* = 0.05 and *** *p* < 0.001). Lozenges (^#^) indicate significance between transgenic treatment groups (placebo vs. RD2RD2: ^###^ *p* < 0.001).

**Figure 2 ijms-22-07066-f002:**
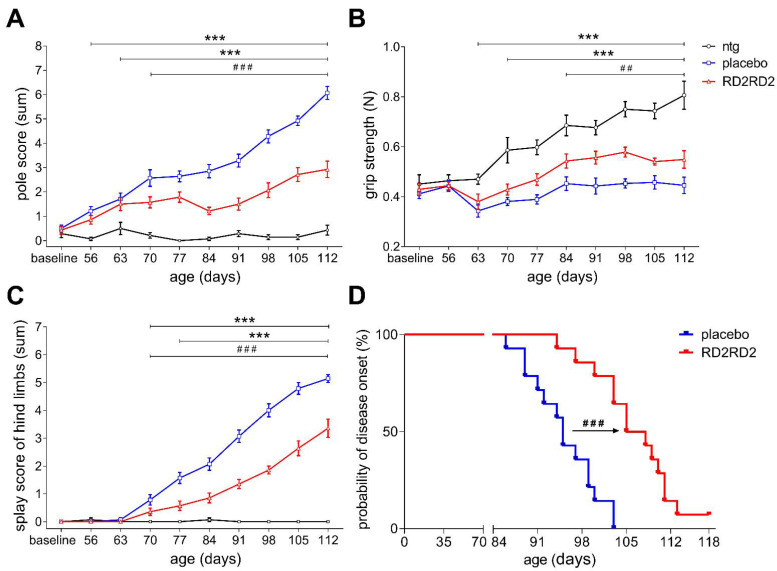
RD2RD2 administration prevented deficits in motor performance and delayed onset of disease in SOD1^G93A^ mice. Motor skills from RD2RD2-treated mice vs. placebo treatment were analysed using different motor tests, i.e., the pole test (**A**). In grip strength and splay reflex test of hind limbs (**B**,**C**) retardation of the motor impairments were measured. Disease onset was assessed using the score of Mead et al. (2011) [45] (**D**). Data is represented as mean ± SEM. Statistical calculations were conducted by two-way RM ANOVA with Fisher’s LSD post hoc analysis, *n* = 14 each group (**A**–**C**) or by Kaplan–Meier survival analysis with log-rank analysis (**D**), placebo = 14 and RD2RD2 *n* = 13. Results of analysis: (**A**) pole test, F(2,351) = 287.81, *p* < 0.001, ntg vs. placebo *p* < 0.002, ntg vs. RD2RD2 *p* < 0.012 and F(1,234) = 99.20, *p* < 0.001, Fisher LSD post hoc analysis, placebo vs. RD2RD2 *p* < 0.001; (**B**) grip strength test, F(2,351) = 54.13, *p* < 0.001, ntg vs. placebo *p* < 0.022, ntg vs. RD2RD2 *p* < 0.002 and F(1,234) = 29.31, *p* < 0.001, Fisher LSD post hoc analysis, placebo vs. RD2RD2 *p* < 0.001 and (**C**) splay reflex test of hind limbs, F(2,351) = 250.93, *p* < 0.001, ntg vs. placebo *p* < 0.001, ntg vs. RD2RD2 *p* < 0.047 and F(1,234) = 79.78, *p* < 0.001, Fisher LSD post hoc analysis, placebo vs. RD2RD2 *p* < 0.001 and (**D**) probability of disease onset (%), log-rank test = 18.31, DF = 1, placebo vs. RD2RD2 *p* < 0.001. Asterisks (*) indicate significance between non-transgenic and transgenic treatment groups (ntg vs. placebo or ntg vs. RD2RD2: *** *p* < 0.001). Lozenges (^#^) indicate significance between transgenic treatment groups (placebo vs. RD2RD2: ^##^ *p* = 0.01 and ^###^ *p* < 0.001).

**Figure 3 ijms-22-07066-f003:**
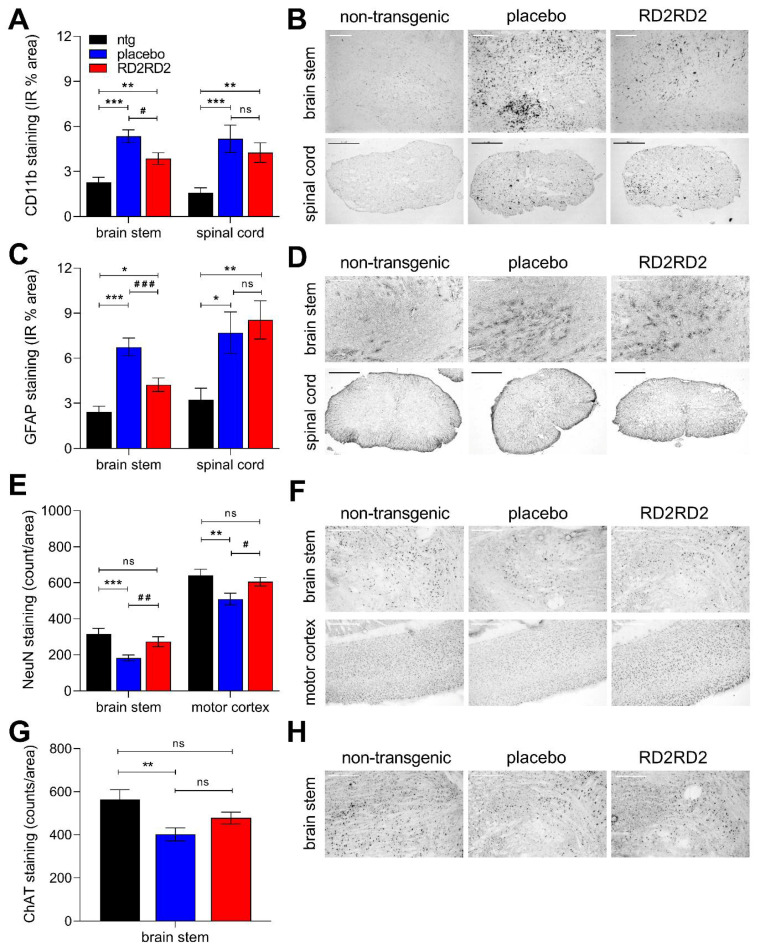
Amelioration of neurodegeneration and neuroinflammation in brain and lumbar spinal cord of SOD1^G93A^ mice by oral treatment with RD2RD2. Investigations of a potential reduction in either gliosis (**A**–**D**) or neuron loss (**E**–**H**) after RD2RD2 treatment were performed using immunohistochemical analysis. Gliosis was analysed by microglia (**A**,**B**) and astrocytes (**C**,**D**) staining in the brain stem (scale bar 200 µm) and lumbar spinal cord (scale bar 500 µm). Preservation of neurons was quantified after mature neuron (**E**,**F**) and motor neuron (**G**,**H**) staining in different areas of the brain (brain stem and motor cortex, scale bar 200 µm). Data is represented as mean ± SEM. Statistical calculations were conducted by one-way ANOVA with Fisher’s LSD post hoc analysis (**A**,**C**,**E**,**G**). Asterisks (*) indicate significance between non-transgenic and transgenic treatment groups (ntg vs. placebo or ntg vs. RD2RD2: * *p* = 0.05, ** *p* = 0.001 and *** *p* < 0.001). Lozenges (^#^) indicate significance between transgenic treatment groups (placebo vs. RD2RD2: ^#^ *p* = 0.05, ^##^ *p* = 0.01 and ^###^ *p* < 0.001). *p*-values of > 0.05 were considered to be statistically not significant (ns). IR: immunoreactivity.

**Figure 4 ijms-22-07066-f004:**
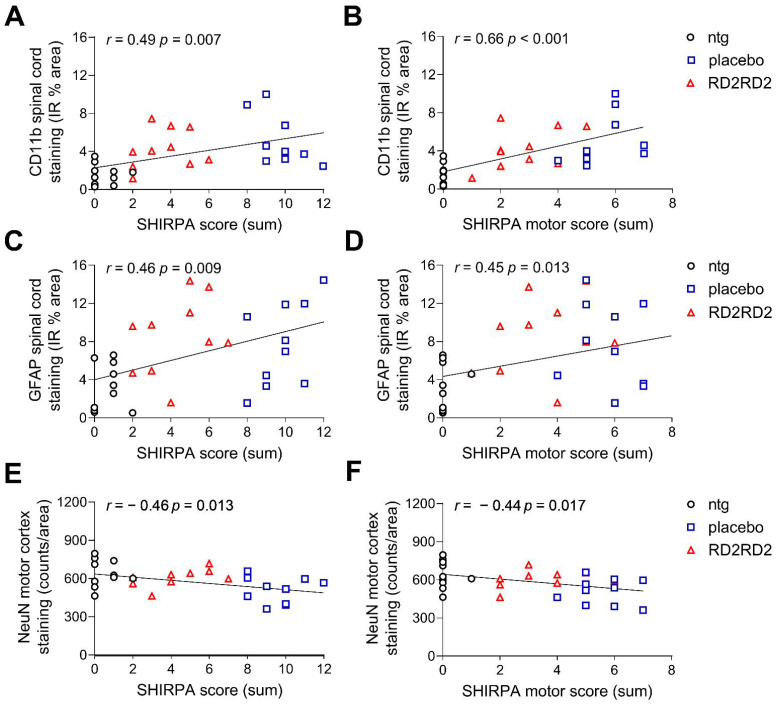
Correlation between histological quantifications and SHIRPA scores over all three groups. Significant correlation was observed between quantified microglia (**A**,**B**) and astrocytes (**C**,**D**) in the spinal cord and the SHIRPA scores of mice. Calculation of the correlation coefficient was performed between quantified mature neurons (**E**,**F**) in the motor cortex and the SHIRPA scores over all three groups. Data were analysed using Pearson correlation coefficient (r). IR: immunoreactivity.

**Figure 5 ijms-22-07066-f005:**
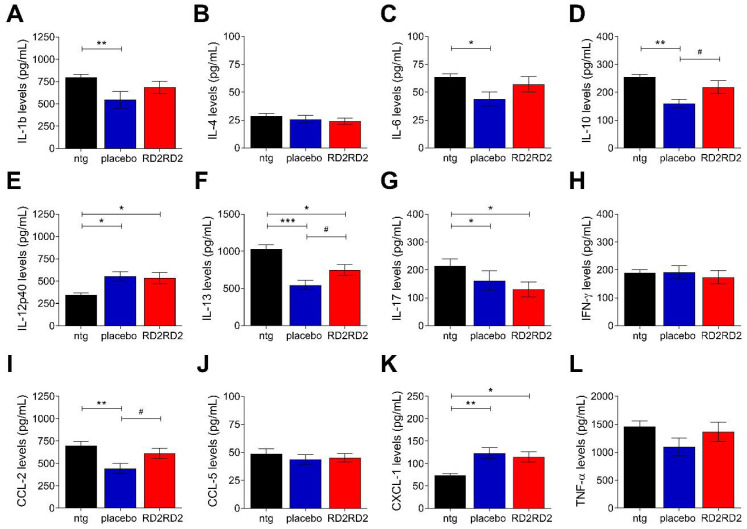
Treatment with RD2RD2 led to differences in several cytokine levels vs. non-transgenic and placebo-treated SOD1^G93A^ mice. A multiplex immunoassay was used to analyse a possible change of inflammatory cytokines at the end of the study. Cytokine concentrations are given in picogram per millilitre (pg/mL). Data is represented as mean ± SEM. Statistical calculations were conducted by one-way ANOVA with Fisher’s LSD post hoc analysis (**A**–**L**), ntg *n* = 7, placebo n = 11 and RD2RD2 *n* = 11. Asterisks (*) indicate significance between non-transgenic and transgenic treatment groups (ntg vs. placebo and ntg vs. RD2RD2: * *p* = 0.05, ** *p* = 0.01 and *** *p* < 0.001). Lozenges (^#^) indicate significance between transgenic treatment groups (placebo vs. RD2RD2: ^#^ *p* = 0.05).

**Figure 6 ijms-22-07066-f006:**
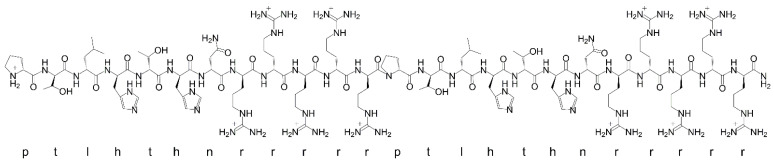
Lewis structure and single letter amino acid code of the d-enantiomeric peptide RD2RD2.

## Data Availability

The datasets used and/or analysed during the current study are available from the corresponding author on reasonable request.

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
