# Peer review of "Oral Treatment with RD2RD2 Impedes Development of Motoric Phenotype and Delays Symptom Onset in SOD1^G93A^ Transgenic Mice"

_ijms, 2021, doi:10.3390/ijms22137066_

Round 1
Reviewer 1 Report
The oral treatment with peptide RD2RD2 impeding development of motoric phenotype and delaying symptom onset in SOD1 transgenic mice is an interesting study.
ALS is a fatal and progressive neurodegenerative disease affecting the motor system and a cure is needed urgently.
The article reads well clearly citating the experiments and the merits of this study and it is a significant contribution to the amyotrophic lateral sclerosis research.
My only suggetion is that the interesting polyarginine structure of the peptide RD2RD2 should be included in the text so the reader can understand better the story.
More than that, I recommend the article to be published as an important contribution to the ALS field.
Reviewer 2 Report
In this article, Post et al. examine the effects of an anti-inflammatory peptide RD2RD2 on the SOD1G93A mouse model of amyotrophic lateral sclerosis (ALS). They find a significant delay of onset and reduction of severity of several ALS-related symptoms in the model mouse, along with reduced gliosis and expression of inflammatory cytokines in plasma. For the most part, their experimental designs and conclusions appear to be sound, however, I am concerned that their use of a single mouse model in only female mice limits the applicability of their study in the broader context of human ALS disease.
Specific comments:
Page 3, paragraph 3, lines 2-4. This sentence is very strangely written and should be reworded
Figure 1. For each panel, in the written results or in the figure legends, it would be helpful to point out whether a higher value represents greater or worse motor function
Page 8, last paragraph. Please indicate whether each correlative relationship is positive or negative
Page 10, paragraph 2, line 2. “efficacy in ALS” should read “efficacy in the SOD1G93A mouse model” or at least “efficacy in ALS model mice”. Caution should always be taken when describing a result observed in non-human models not to assume the result will be the same in the human disease.
Page 10, paragraph 6, line 3. “significant survival of neurons” should read “significant increase in the survival of neurons”
Somewhere in the discussion section, the authors need to address the limitations of this study and in the choice of mouse model. First, familial ALS, which the SOD1G93A mouse model mimics, is only around 10% of all ALS cases, and only a subset are caused by SOD1 mutation. Ideally, studies would be performed in parallel in other mouse models, such as TDP-43 and C9orf72 mutants. Given the expense of large-scale mouse studies, I understand why this was not done, however, it should be addressed in the discussion section. In addition, it is a huge weakness of this study that it was only performed in female mice. ALS has a differential disease rate in men and women, and drug responses between female and male mice may not present in the same way. The optimal study design would incorporate both male and female mice and analyze them both separately and together in order to make conclusions. I am not asking the authors to repeat the study, but some language needs to be added into the Discussion section addressing the caveats and limitations of the current study.
